# Research on a Two-Layer Optimal Dispatching Method Considering the Mutual Aid of Peak Regulating Resources among Regional Power Grids

**Tianmeng Yang [1], Suhua Lou [2,*], Meng Zhang [3], Yanchun Li [1], Wei Feng [1] and Jicheng Liu [1]**

[1] Northeast Branch of State Grid Corporation of China, Shenyang 110180, China; 18240369315@163.com (T.Y.); mty18842620317@163.com (Y.L.); 13971415920@163.com (W.F.); m202271914@hust.edu.cn (J.L.)
[2] State Key Laboratory of Advanced Electromagnetic Engineering and Technology, Wuhan 430074, China
[3] China Energy Engineering Group Liaoning Electric Power Design Institute Co., Ltd., Shenyang 110180, China; mzhang4947@ceec.net.cn
* Correspondence: shlou@mail.hust.edu.cn; Tel.: +86-132-9661-3265

**Abstract:** Since the power generation structures and load characteristics in each province in China are quite different, the distribution of peak regulating resources and demands are extremely imbalance. Restricted by a low power marketization degree, peak regulating resource shortages, and transmission channel blocks, the efficient utilization of new energy is facing greater pressures. In order to improve the mutual aid in regional power grids and to obtain more precise simulation results, this paper proposes a two-layer optimization dispatching model, considering the mutual aid of peak regulation resources between each province. It determines the optimal startup mode and the units' power output in each province and obtains the power output arrangements for all the units and the technical and economic indicators. The model and the solution method are original and innovative. And it effectively solved the unequal distribution problem between the peak regulating demands and resources of each provincial power grid. Finally, taking an actual regional power grid in China as an example, the simulation results show that the proposed model can significantly improve the utilization rate of new energy, which verifies the effectiveness and feasibility of the proposed model and methods presented in this paper.

**Keywords:** regional power grids; optimal dispatching; peak regulation resource mutual assistance; peak regulation capability; two-layer optimization

## 1. Introduction

With the proposal of "carbon peak and carbon neutral", power generation structures are turning to a low-carbon transformation, and renewable energy power is developing rapidly in China [1–3]. In order to make full use of renewable energy, the government has put forward high requirements for the utilization rate [4–6]. Therefore, how to plan the power grid's development and optimize the operation modes, and how to make full use of the limited, flexible regulating resources are critical problems. What is more, how to maximize the utilization rate of renewable energy and reduce the peak regulating costs have become critical problems, which need to be solved for the current renewable energy development process in China [7–10].

Currently, there are papers that have studied the optimal dispatching operation methods based on considering the full utilization of peak regulating resources, but the research objects have been provincial power grids, and these are not representative. In [11], a multi-objective optimal scheduling model, considering the economy and flexibility of deep peak regulation, is constructed, but the power grid under study is not representative. In [12,13], by establishing the peak regulation right transaction of wind power and thermal power units, a combined thermoelectric economic dispatching model is established. It

can stimulate the predisposition of thermoelectric units to participate in peak regulation, which can improve the wind power utilization rate. In [14,15], based on a peak regulating energy consumption cost model for the different stages of thermal power units, an economic dispatching model, which gives priority to wind power utilization, is established. In [16,17], based on the Manson–Coffin formula, a thermal power unit loss cost model with variable loads is established. And based on the full acceptance of wind power, an economic dispatching model based on a hierarchical deep peak regulation is established, which aims to minimize the total power generation costs.

In the studies of the optimization dispatching method in [18,19], the feasibility of peak regulation between regional power grids is verified. In [20,21], to solve the possible shortage problems of peak regulating resources, a transaction mechanism for peak regulation auxiliary service between the regional power grids is proposed, and a clearing pricing model is first constructed. In [22,23], based on an analysis of the power source and load characteristics of multiple regions, a typical peak–valley mutual aid operation scheme is given, and the typical operation mode and economic evaluation results are optimized. In [24], a two-stage iterative power exchange optimization method considering the marginal power generation costs is put forward. However, this method only aims at obtaining the minimum operation costs and cannot fully use regional peak regulating resources. In [25], an environmental economy dispatching model, considering the inter-provincial power balance of a regional power grid, is established. However, only the actual power limitation is taken into consideration, and the influence of the inter-provincial connection lines' actual electricity trading is not taken into consideration. In [26], by studying the different power sources' peak regulating characteristics, an optimal dispatching method considering peak regulating resources is studied. However, only the deep peak regulating operation costs of thermal power units are considered, and the influence of other peak regulating power sources is not considered.

In summary, peak regulating resources can be exchanged through inter-provincial transmission lines between provincial power grids in China, so as to make full use of peak regulating resources in the regional power grid. Based on this, this paper analyzes the peak regulation characteristics of different power sources, fully considers the differences between the surplus and gap of peak regulation power in the different provinces, and puts forward a two-layer optimization dispatching model with the lowest operation costs and highest utilization rate of renewable energy for a regional power grid. The upper-layer optimization model's objective function is the lowest operation costs for a regional power grid, and the lower-layer optimization model's objective function is the highest utilization rate of renewable energy. Based on this two-layer optimization model, the inter-provincial transmission curve with the lowest costs and the highest renewable power utilization rate can be re-optimized. And the optimized operation results for the whole regional power grid can be obtained. And last, taking an actual regional power grid as an example, the feasibility and high efficiency of the proposed model and optimization method are verified from the simulation results.

## 2. System Peak Regulating Resource Analysis

### 2.1. Analysis of Different Power Supplies

The peak regulation scale for active power and the quantity of electricity is large in China. Now, the main resources that can effectively participate in peak regulation are coal-fired units, gas units, cascade hydro-power units (non-runoff), pumped storage, and other types of large-capacity power storage. Among them, coal-fired units are non-stop peaking units, and gas units and cascade hydro-power units can be used as stop peaking units, thanks to their rapid start and stop characteristics [24,25]. The peak regulating characteristics of each power supply are shown in Table 1.

**Table 1.** Peak regulating characteristics of different power supplies.

| Power Supply | Peak Regulating Ability |
|---|---|
| Coal-fired units | Conventional thermal units: 40~50% |
| | After flexibility transformation: 55~70% |
| Gas units | Have shutdown ability, peak regulation ability: 100% |
| Cascade hydro-power unit (non-runoff) | Have shutdown ability, peak regulation ability: 100% |
| Pump storage units | 160% |
| Other energy storage units | 160% |
| Small hydro-power stations (radial flow) | 0% |
| Wind | 0% |
| Photovoltaic | 0% |

*2.2. Analysis of Coal-Fired Thermal Units' Energy Consumption Costs*

Considering the rapid development of renewable energy, the function of coal-fired units has gradually changed from the main electricity source to regulating the power supply. When the coal-fired thermal units change to a regulating function, all the units' operating states change frequently. And the characteristics of the technical and economic costs also significantly change.

Considering the operating states and the energy consumption characteristics of coal-fired units, the coal-fired units' peak regulating processes can be divided into three stages: a basic peak regulation stage (RPR), a non-oil supply peak regulation stage (DPR), and an oil supply peak regulation stage (DPRO). The characteristic curves for the energy consumption costs during each peak regulating period are shown in Figure 1. $P_{\text{max}}^{G}$ represents the rated power output of the coal-fired unit, $P_{\text{a}}^{G}$ represents the minimum technical power output of the RPR stage, $P_{\text{b}}^{G}$ represents the minimum technical power output of the DPR stage, and $P_{\text{c}}^{G}$ represents the minimum technical power output of the DPRO stage.

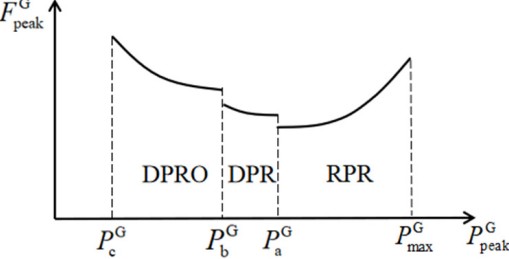

**Figure 1.** Peak regulating costs of the fuel units.

In recent years, thanks to the technical progress of thermal units, the minimum stable combustion load of the 200 MW and 300 MW units can be reduced to 45% of rated power after flexibility transformation. And the minimum stable combustion load of the 600 MW and 1000 MW units can be reduced to 30~35% after flexibility transformation. The formula of thermal power units' energy consumption costs is shown below.

$$F_{\text{peak}}^{G} = \begin{cases} F_{\text{coal}}(P_{\text{peak}}^{G}) & P_{\text{a}}^{G} < P_{\text{peak}}^{G} \leq P_{\text{a}}^{\text{max}} \\ F_{\text{coal}}(P_{\text{peak}}^{G}) + F_{\text{loss}}(P_{\text{peak}}^{G}) & P_{\text{b}}^{G} < P_{\text{peak}}^{G} \leq P_{\text{a}}^{G} \\ F_{\text{coal}}(P_{\text{peak}}^{G}) + F_{\text{loss}}(P_{\text{peak}}^{G}) + F_{\text{oil}}(P_{\text{peak}}^{G}) \\ + F_{\text{en}}(P_{\text{peak}}^{G}) & P_{\text{c}}^{G} < P_{\text{peak}}^{G} \leq P_{\text{b}}^{G} \end{cases} \tag{1}$$

In the formula, $F_{\text{coal}}(P_{\text{peak}}^{G})$ represents the thermal power units' operating coal consumption costs, which is described by the second-order consumption model in (2).

$$F_{\text{coal}}(P_{\text{peak}}^{G}) = k_{\text{coal}} \cdot (\alpha P_{\text{peak}}^{G}{}^{2} + \beta P_{\text{peak}}^{G} + \gamma) \tag{2}$$

In the formula, $k_{\text{coal}}$ represents the coal price coefficient, and $\alpha$, $\beta$, $\gamma$ represent the consumption characteristic coefficients of thermal units, respectively.

$F_{\text{loss}}(P_{\text{peak}}^{\text{G}})$ represents the unit loss costs caused by the excessive thermal stress in the DPR and DPRO stages. The unit loss costs are mainly determined by the cracking cycle of the rotor, as shown in (3).

$$F_{\text{loss}}(P_{\text{peak}}^{\text{G}}) = \frac{1}{2N_{\text{T}}(P_{\text{peak}}^{\text{G}})}\tau k_{\text{unit}} \tag{3}$$

In the formula, $N_{\text{T}}(P_{\text{peak}}^{\text{G}})$ represents the cracking cycle of the rotor, which is interrelated with the power output of units. $\tau$ represents the actual operating loss coefficient of the thermal power units, and the value in the DPRO stage is larger than in the DPR stage. $K_{\text{unit}}$ represents the investing cost coefficient of thermal power units.

$F_{\text{oil}}(P_{\text{peak}}^{\text{G}})$ represents the fuel consumption costs in the DPRO stage, which is mainly determined by fuel consumption quantity and oil price in the DPRO stage, as shown in (4).

$$F_{\text{oil}}(P_{\text{peak}}^{\text{G}}) = k_{\text{oil}} \cdot E_{\text{oil}} \tag{4}$$

In the formula, $k_{\text{oil}}$ represents oil price, and $E_{\text{oil}}$ represents fuel consumption quantity in the DPRO stage.

$F_{\text{en}}(P_{\text{peak}}^{\text{G}})$ represents additional environmental punishment fees caused by pollutants' excessive discharge in the DPRO stage, which is shown in (5).

$$F_{\text{en}}(P_{\text{peak}}^{\text{G}}) = k_{\text{punish}} \cdot \lambda_{\text{S}} \tag{5}$$

In the formula, $k_{\text{punish}}$ represents the penalty cost coefficient of pollutant discharge, and $\lambda_{\text{S}}$ represents the pollutant emissions.

### 2.3. Analysis of Pumped Storage Units' Peak Regulating Costs

As a main tool for peak regulating, the pumped storage units have functions including peak load shifting and valley filling, frequency regulating and phase regulating, spinning reserve, and black start. On the one hand, pumped storage units have flexible operating modes with reversible pump turbine units; on the other hand, limited by different operating conditions, pumped storage units have specific operating characteristics and constraints.

The pumped storage unit works the same as the conventional generator in the generating state. In the pumping state, it absorbs power from the power grid. And the operating costs generated by pumped storage units are mainly generated by the conversion between pumping and generating states, which consist of generators' and motors' start-up costs.

$$F_{\text{peak},j,t}^{\text{PH}} = F_{j,t}^{\text{PH}-\text{G}} + F_{j,t}^{\text{PH}-\text{P}} \tag{6}$$

In the formula, $F_{j,t}^{\text{PH}-\text{G}}$ represents the starting costs at the $t$th period of the $j$th pumped storage unit working in the generating state, and $F_{j,t}^{\text{PH}-\text{P}}$ represents the costs in the pumping state.

## 3. Two-Layer Optimization Dispatching Model of the Regional Power Grid

### 3.1. Introduction of the Model

Considering the differences of the power generation structures and load characteristics of each provincial power grid, the distribution of peak regulating demands is extremely imbalanced. Therefore, the regional peak regulating resources should be fully utilized, which can improve the new energy utilization rate of the regional power grid.

Thus, based on the provincial power balance and dispatching management principles in China, a two-layer optimization dispatching model of the regional power grids is proposed in this paper. The model aims to divide the optimal dispatching problem of the regional power grid into two parts: the units' optimal dispatching of each province and the optimization of inter-provincial power transmission plan curves. This two-layer model is extensible and it can be applied into any regional power grid. The flow chart of the two-layer optimization method is shown in Figure 2.

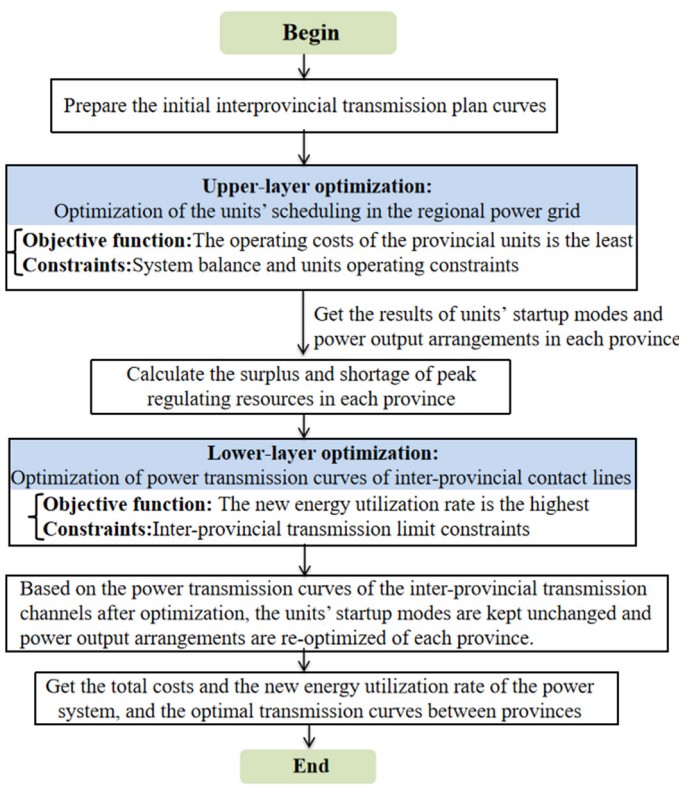

**Figure 2.** Two-layer optimization dispatching process of regional power grids.

### 3.2. Units' Optimization Dispatching Model in Each Province

#### 3.2.1. Objective Function

The units' optimization dispatching model is for optimizing the startup modes and power output in each province on the basis of the given power transmission plan curves, so that the total operation costs in each province are the lowest. Taking one province as an example, the objective function aims to obtain the minimum operation costs, as shown in Equation (7).

$$\min F = \min(\alpha_1 \cdot F_{\text{oper}} + \alpha_2 \cdot F_{\text{dep}}) \tag{7}$$

In the formula, $\alpha_1$ and $\alpha_2$ represent the weight coefficients of operation costs and new energy abandonment penalty costs, respectively. $F_{\text{oper}}$ represents the operating costs of all units in the system, including the operating costs of thermal power units and pumped storage units. $F_{\text{dep}}$ represents the penalty costs of new energy abandonment.

$$\begin{cases} F_{\text{oper}} = F_{\text{peak}}^{\text{G}} + F_{\text{peak}}^{\text{PH}} \\ F_{\text{peak}}^{\text{G}} = \sum\limits_{t=1}^{T} \sum\limits_{i=1}^{N_{\text{G}}} F_{\text{peak},i,t}^{\text{G}} \\ F_{\text{peak}}^{\text{PH}} = \sum\limits_{t=1}^{T} \sum\limits_{j=1}^{N_{\text{PH}}} F_{\text{peak},j,t}^{\text{PH}} \end{cases} \tag{8}$$

$$F_{\text{dep}} = k_{\text{dep}} \cdot E_{\text{dep}} \tag{9}$$

In the formula, $F_{\text{peak},i,t}^{\text{G}}$ and $F_{\text{peak},i,t}^{\text{PH}}$ represent the peak regulating costs of the $i$th thermal unit and $j$th pumped storage unit, respectively, $k_{\text{dep}}$ represents the penalty cost coefficient of abandoned power, and $E_{\text{dep}}$ represents abandoned electricity of new energy.

As shown above, the peak regulating costs of thermal power units are mainly obtained according to Formula (1), and the peak regulating costs of pumped storage units are mainly obtained according to Formula (6).

### 3.2.2. Constraints

System Power Balance Constraints

$$P_t^{\mathrm{L}} + P_t^{\mathrm{C}} = P_t^{\mathrm{N}} + \sum_{i=1}^{N_{\mathrm{G}}} P_{\mathrm{peak},i,t}^{\mathrm{G}} + \sum_{j=1}^{N_{\mathrm{PH}}} P_{j,t}^{\mathrm{PH}} - P_{\mathrm{dep},t} \tag{10}$$

In the formula, $P_t^{\mathrm{L}}$ represents the load level at the $t$th period; $P_t^{\mathrm{C}}$ represents the reserve capacity at the $t$th period; $P_t^{\mathrm{N}}$ represents the power output of new energy at the $t$th period; $P_{\mathrm{peak},i,t}^{\mathrm{G}}$ represents the power output of the $i$th thermal unit at the $t$th period; $P_{j,t}^{\mathrm{PH}}$ represents the power output of the $j$th pump storage unit at the $t$th period, with a positive value representing the generating state and negative value representing the pumping state; $P_{\mathrm{dep},t}$ represents the abandoned power output at the $t$th period.

Thermal Power Unit Climbing Constraints

$$\mathrm{DP}_i^{\mathrm{G}} \leq P_{\mathrm{peak}.i,t}^{\mathrm{G}} - P_{\mathrm{peak},i,t-1}^{\mathrm{G}} \leq \mathrm{UP}_i^{\mathrm{G}} \tag{11}$$

In the formula, $\mathrm{DP}_i^{\mathrm{G}}$ and $\mathrm{UP}_i^{\mathrm{G}}$ represent the climbing up and down limitations of the thermal unit, respectively.

Pumped Storage Operation Constraints

According to the actual operating restriction of pumped storage units, the pumped storage unit pumps with rated power in the pumping state. And pumped power generation should meet the storage capacity constraints. Its power output and storage capacity constraints are shown below.

$$P_{j,t}^{\mathrm{PH-P}} = u_{j,t}^{\mathrm{PH-P}} \cdot P_{j,\max}^{\mathrm{PH}} \tag{12}$$

$$0 \leq P_{j,t}^{\mathrm{PH-G}} \leq u_{j,t}^{\mathrm{PH-G}} P_{j,\max}^{\mathrm{PH}} \tag{13}$$

$$E_{\mathrm{PH,min}} \leq E_{\mathrm{PH},t} \leq E_{\mathrm{PH,max}} \tag{14}$$

$$E_{\mathrm{PH},t} = E_{\mathrm{PH},t-1} + \Delta T \cdot \left( \sum_{j=1}^{N_{\mathrm{PH}}} \eta_{\mathrm{PH-P}} \cdot P_{j,t}^{\mathrm{PH-P}} - \sum_{j=1}^{N_{\mathrm{PH}}} \frac{P_{j,t}^{\mathrm{PH-G}}}{\eta_{\mathrm{PH-G}}} \right) \tag{15}$$

In the formula, $P_{j,\max}^{\mathrm{PH}}$ represents the rated power output of the $j$th pumped storage unit; $u_{j,t}^{\mathrm{PH-P}}$ and $u_{j,t}^{\mathrm{PH-G}}$ represent the pumping and generating state variables, respectively, which satisfy $u_{j,t}^{\mathrm{PH-P}} + u_{j,t}^{\mathrm{PH-G}} \leq 1$; $P_{j,t}^{\mathrm{PH-P}}$ and $P_{j,t}^{\mathrm{PH-G}}$ represent the pumping power and generating power of the $j$th unit at the $t$th period, respectively. $E_{\mathrm{PH,max}}$ represents the rated storage capacity of the pumped storage station; $E_{\mathrm{PH,min}}$ represents the minimum storage capacity of the pumped storage station; $\eta_{\mathrm{PH-P}}$ and $\eta_{\mathrm{PH-G}}$ represent the pumping and generating efficiency of pumped storage units.

Upper and Lower Limits of Thermal Power Units

$$P_{\mathrm{c},i}^{\mathrm{G}} \leq P_{\mathrm{peak},i,t}^{\mathrm{G}} \leq P_{i,\max}^{\mathrm{G}} \tag{16}$$

In the formula, $P_{\mathrm{c},i}^{\mathrm{G}}$ represents the lowest power output limit of the $i$th thermal power unit.

### 3.3. Analysis of Peak Regulation Amplitude and Gap between Provinces in the Regional Power Grid

By solving the provincial unit optimization model, the startup plan and power output arrangements can be obtained. They are calculated with the initial inter-provincial power transmission curves. Thus, we calculate the amplitude and gap indexes of peak regulating

resources in each province, which can provide the basis for optimizing the transmission curves between provincial power grids.

$$
\begin{cases}
P_t^{\text{margin}} = \sum_{i=1}^{N_{\text{G}}} \left( P_{\text{peak},i,t}^{\text{G}} - P_{\text{c},i}^{\text{G}} \right) \\
P_t^{\text{insuff}} = P_{\text{dep},t}
\end{cases} \tag{17}
$$

In the formula, $P_t^{\text{margin}}$ and $P_t^{\text{insuff}}$ represent the amplitude and gap of peak regulation resource at the $t$th period, respectively.

*3.4. Optimization Scheduling Model of Inter-Provincial Transmission Curves*

Based on the peak regulation amplitude and gap in each provincial power grid calculated in Section 3.3, the initial inter-provincial transmission curves should be optimized and adjusted. Based on the optimized transmission curves, we can obtain the highest new energy utilization rate of the whole regional power grid. And the peak regulating resources will be called up in a more balanced manner.

3.4.1. Objective Function

The objective function of the inter-provincial transmission curve optimization model mainly considers two aspects; these are the highest new energy utilization rate and the lowest system peak regulation costs. The objective function is shown below.

$$
\min E_{\text{dep}} = \min\left( \beta_1 \cdot k_{\text{dep}} \cdot \sum_{t=1}^{T} P_{\text{dep},t}' + \beta_2 \cdot k_{\text{peak}} \cdot \sum_{l=1}^{N_{\text{L}}} \sum_{t=1}^{T} \Delta P_{l,t}^{\text{D}} \right) \tag{18}
$$

In the formula, $\beta_1$ and $\beta_2$ represent the weight coefficients of new energy abandonment penalty costs and peak regulating costs; $k_{\text{peak}}$ represents the coefficient of system peak regulating costs; $P_{\text{dep},t}'$ represents the abandoned new energy power output at the $t$th period after optimizing the inter-provincial transmission curves; $\Delta P_{l,t}^{\text{D}}$ represents power differences of the $l$th inter-provincial transmission lines before and after optimization at the $t$th period.

Thus, the calculation formulas of the abandoned new energy power output are as follows:

$$
\begin{cases}
P_{\text{dep},t}' = P_{\text{dep},t} + \sum_{l=1}^{N_{\text{L}}} \Delta P_{l,t}^{\text{D}} \\
\sum_{l=1}^{N_{\text{L}}} \Delta P_{l,t}^{\text{D}} \leq P_t^{\text{insuff}}
\end{cases} \tag{19}
$$

3.4.2. Constraints
Transmission Limit Constraints

$$
\begin{cases}
P_{l,t}'^{\text{D}} = P_{l,t}^{\text{D}} + \Delta P_{l,t}^{\text{D}} \\
P_{l,t}'^{\text{D}} \leq P_{l,\text{N}}^{\text{Pos}}, P_{l,t}'^{\text{D}} \geq 0 \\
P_{l,t}'^{\text{D}} \geq P_{l,\text{N}}^{\text{Neg}}, P_{l,t}'^{\text{D}} < 0
\end{cases} \tag{20}
$$

In the formula, $P_{l,t}^{\text{D}}$ and $P_{l,t}'^{\text{D}}$ represent the transmission power of the $l$th channel at the $t$th period before and after optimization, respectively, while $P_{l,\text{N}}^{\text{Pos}}$ and $P_{l,\text{N}}^{\text{Neg}}$ represent the transmission limitation of the $l$th connection channel in two directions, respectively.

Constraints of Transmission Power Limitation

$$
E_l^{\text{D}} = \sum_{t=1}^{T} \sum_{l=1}^{N_{\text{L}}} P_{l,t}'^{\text{D}} \tag{21}
$$

$$
E_l^{\text{min}} \leq E_l^{\text{D}} \leq E_l^{\text{max}} \tag{22}
$$

In the formula, $E_l^D$ represents the transmission electricity of the *l*th transmission channel after optimization; $E_l^{max}$ and $E_l^{min}$ represent the upper and lower transmission electricity limitation of the *l*th transmission channel.

## 4. Case Study

In order to fully verify the effectiveness and feasibility of the proposed model and method, two example power systems are presented for simulation and analysis. They are the IEEE RTS-96 test system and a regional power grid in China, respectively. Since the two-layer model is non-linear, all optimization processes mainly depend on Matlab Software R2016a with CPLEX and the Yalmip optimization engine. Considering the uncertainty of new energy power, the clustering algorithm is used to gather typical scenarios to represent all historical curves.

### 4.1. Example System Analysis Based on IEEE RTS-96

#### 4.1.1. Overview of the Example System

The example system is modified from the IEEE RTS-96 system, and all parameters of the system are based on the original system. It is divided into three sub-regional power grids. Each region is connected through DC transmission channels. The load characteristics of the three sub-regions are shown in Figure 3. The load level of region 1 and region 2 is lower than that of region 3, and it has the smallest peak–valley difference. The peak regulating abilities of thermal units are shown in Table 2. The wind power capacities are 1300 MW, and the wind power output characteristics in each region are shown in Figure 4. The peak regulating characteristics of wind power output in region 1 are positive, which can reduce the peak regulating pressure, while it is negative in region 2, which aggravates the peak regulating pressure. The characteristics of wind power output in region 3 are stable, which has little influence on peak regulation.

**Table 2.** Peak regulating capacities of thermal units in each sub-region.

| Region | Units Capacities/MW | Wind Power Capacities/MW | Peak Regulating Capacities | Peak Regulating Capacities without Oil | Peak Regulating Capacities with Oil |
|---|---|---|---|---|---|
| Region 1 | 29,410 | 1300 | 50% | 40% | 35% |
| Region 2 | 29,410 | 1300 | 50% | 45% | 40% |
| Region 3 | 21,100 | 1300 | 50% | 43% | 38% |

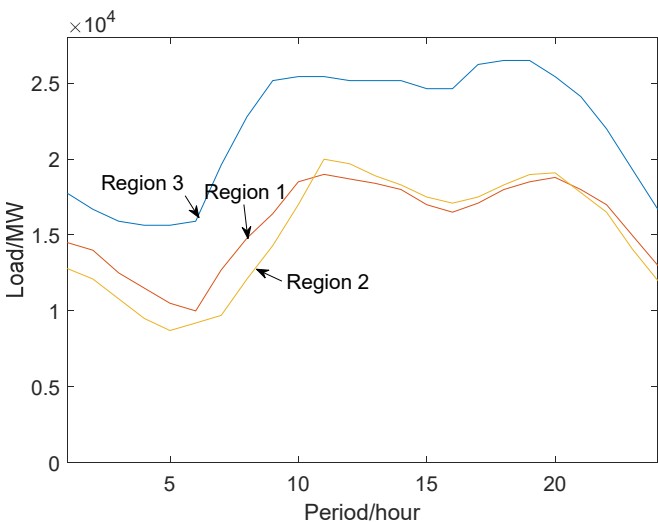

**Figure 3.** Load characteristics of regional power grid.

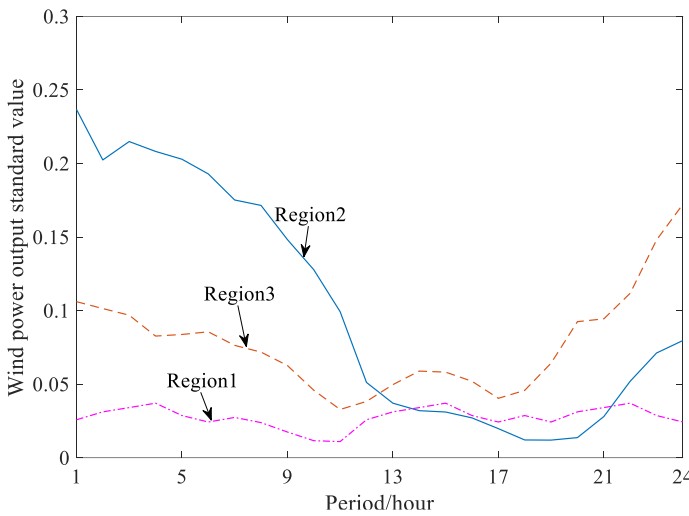

**Figure 4.** Wind power characteristics of regional power grid.

The weight coefficients $\alpha_1$ and $\alpha_2$ are 0.7 and 0.3, and $\beta_1$ and $\beta_2$ are 0.8 and 0.2.

### 4.1.2. Analytical Calculation

Regions 1 and 2 are the main power delivery areas with surplus unit capacities and small load level, while region 3 is the main power-receiving area with insufficient unit capacities and large load level. The initial transmission curves between regions take the constant power transmission mode. The results of operating costs in each region before and after optimization are shown in Table 3, and the regional daily transmission curves before and after optimization are shown in Figure 5.

**Table 3.** Operation costs of each region before and after inter-provincial transmission curve optimization.

| Region | Before Optimization | | | After Optimization | | |
|---|---|---|---|---|---|---|
| | Total Costs/USD 100 Million | Operation Costs/USD 100 Million | Abandon Punishment Costs/USD 100 Million | Total Costs/USD 100 Million | Operation Costs/USD 100 Million | Abandon Punishment Costs/USD 100 Million |
| 1 | 0.28 | 0.23 | 0.05 | 0.24 | 0.22 | 0.02 |
| 2 | 0.27 | 0.23 | 0.04 | 0.25 | 0.23 | 0.02 |
| 3 | 0.25 | 0.2 | 0.05 | 0.22 | 0.19 | 0.03 |
| Total | 0.8 | 0.66 | 0.14 | 0.71 | 0.64 | 0.07 |

As can be seen in Figure 5, the power transmission curves of regions 1 and 2 after optimization show great differences. The main reason is that there exist extreme differences between peak regulating capacities and demands in different regions. Since the load peak and valley differences in region 1 are small, the peak regulation demands are the minimum. While the load peak and valley differences in region 2 are large, the peak regulation demands are the maximum, and the peak regulating support ability to other regions is the minimum. Therefore, the optimized transmission curves can fully call up peak regulating resources in the three regions. During low-load periods, region 2 transfers power to region 3, and region 1 transfers power mainly during high-load periods. Thus, it can reduce system operating costs and power abandonment penalty costs, and promote the operating efficiency of each regional power grid.

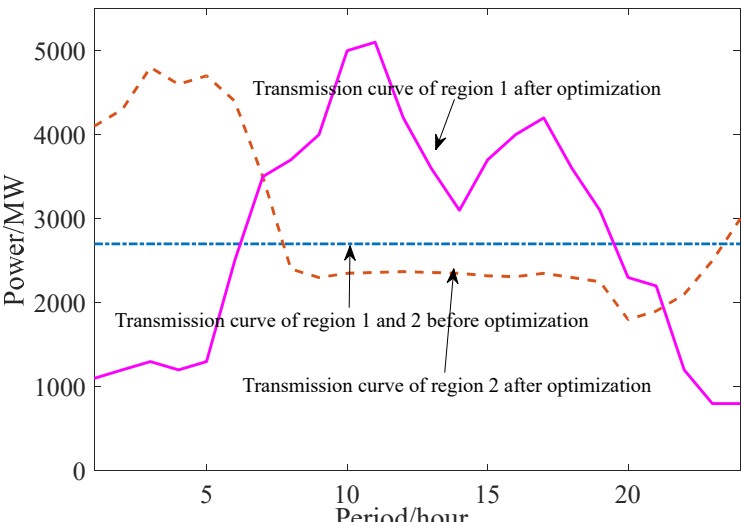

**Figure 5.** Daily power transmission curves between each regional power grid before and after inter-provincial transmission curve optimization.

### 4.2. Example System Analysis Based on an Actual Regional Power Grid

#### 4.2.1. Overview of the Regional System

In this part, an actual regional power grid in China is taken as an example for analysis and calculation, and this region consists of four provincial power grids. All the parameters of units and load curves are based on the actual power system. In this region, provinces A, B, and C have surplus electricity and province D mainly receives electricity from B and C. The actual power flows are shown in Figure 6. And in actual operation, the power flow direction is mainly opposite to what is shown in Figure 6.

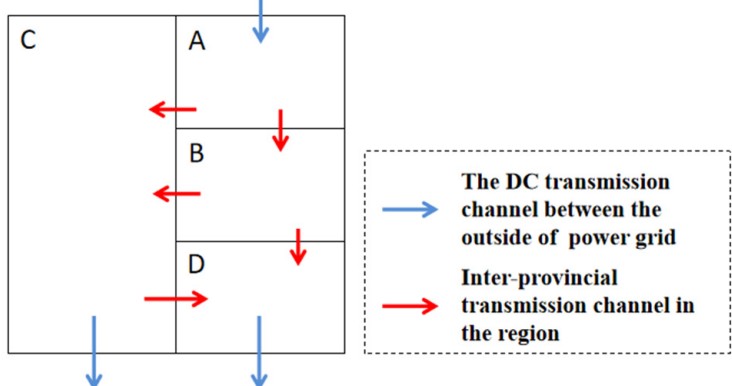

**Figure 6.** Schematic diagram of power transmission and reception between each province.

Since this region is located at high latitude in China, the load demands and the peak regulating demands are large in winter. Therefore, taking winter load characteristics as an example, the load characteristics of the four provinces are shown in Figure 7. The load demands of provinces A, B, and C are small and the load peak valley differences of province B are the smallest. The power output characteristics of each province are shown in Figure 8.

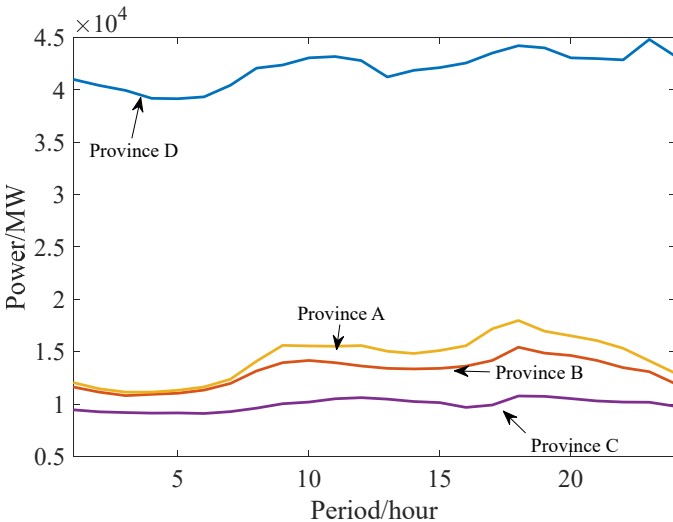

**Figure 7.** Load characteristics of each province.

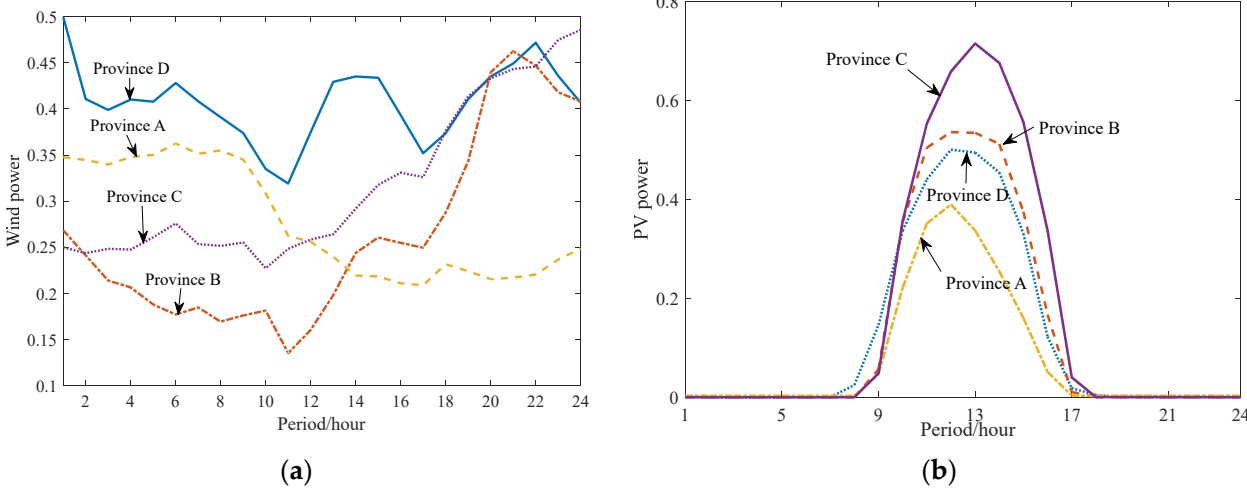

(**a**)

(**b**)

**Figure 8.** (**a**) Wind power output characteristics of each province in regional power grid. (**b**) PV power output characteristics of each province in regional power grid.

### 4.2.2. Analytical Calculation

Before optimization, the DC power transmission curves outside the region and the initial transmission curves between provinces in the region are based on the historical annual transmission curves. The operating cost results before and after optimization are shown in Table 4. The regional daily transmission curves before and after optimization are shown in Figure 9.

**Table 4.** Operation costs of each regional power grid before and after inter-provincial transmission plan curve optimization.

| Province | Before Optimization | | | | After Optimization | | | |
|---|---|---|---|---|---|---|---|---|
| | Total Costs/USD 100 Million | Operation Costs/USD 100 Million | Abandon Punishment Costs/USD 100 Million | New Energy Utilization Rate% | Total Costs/USD 100 Million | Operation Costs/USD 100 Million | Abandon Punishment Costs/USD 100 Million | New Energy Utilization Rate% |
| A | 0.46 | 0.396 | 0.064 | 87.9 | 0.419 | 0.377 | 0.042 | 89.9 |
| B | 0.545 | 0.452 | 0.092 | 84.8 | 0.495 | 0.433 | 0.061 | 88.3 |
| C | 0.455 | 0.335 | 0.121 | 89.6 | 0.423 | 0.315 | 0.108 | 90.7 |
| D | 0.653 | 0.599 | 0.053 | 93.3 | 0.617 | 0.565 | 0.052 | 93.5 |
| Total | 2.113 | 1.782 | 0.33 | 87.7 | 1.954 | 1.69 | 0.263 | 90.9 |

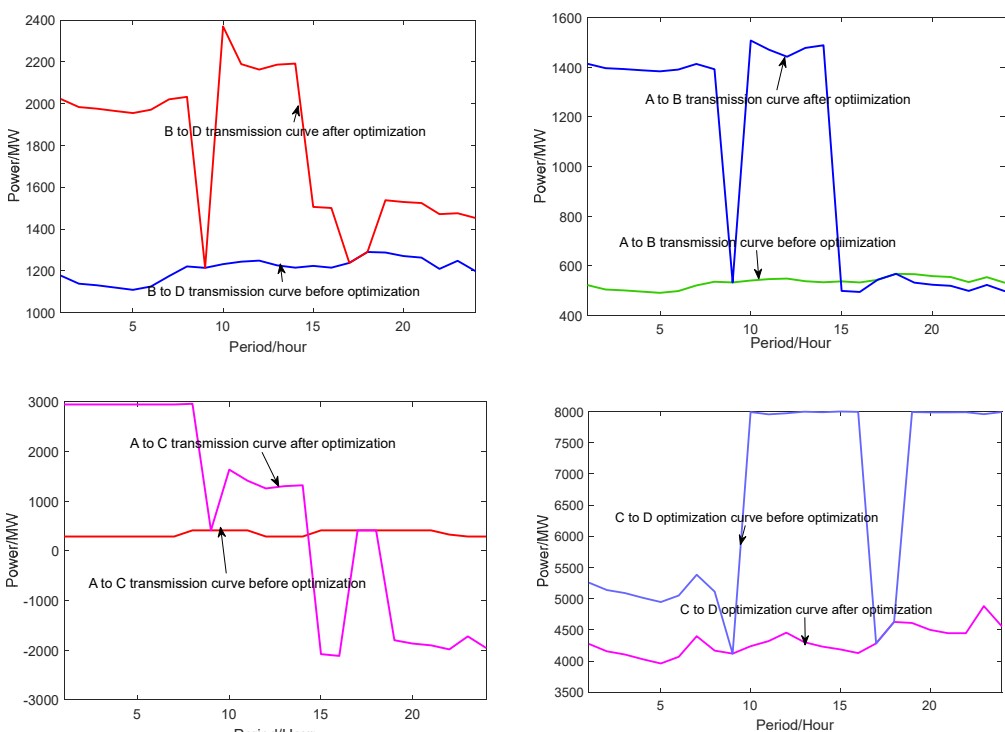

**Figure 9.** Daily power transmission curves of each regional power grid before and after transmission curve optimization.

As can be seen from Figure 9, the differences in the transmission curves between provinces C and D and between provinces B and D before and after optimization are the largest. The main reason is that the peak regulating capacities and amplitude of province D are the largest. After optimizing the transmission curves between C and D and between B and D, the surplus peak regulation resources can be supported to B and C, which can maximize the new energy utilization rate of the whole regional power grid.

According to the calculation results in Table 4, after the unit startup optimization of the whole system and transmission curve optimization between provinces, the total costs of the regional power grid decrease by 7.6%, the operation costs decrease by 5.2%, and the new energy power abandonment penalty costs decrease by 20.4%. After optimization, the utilization rate of new energy in the whole region increases by 3.2%. And the utilization rate of new energy in province B has the maximum improvement (3.5%).

### 4.3. Comparison and Analysis

In order to further verify the effectiveness and feasibility of the proposed optimization model and algorithm, the traditional optimization method based on equal peak regulating rate was used for comparison. In the traditional equal peak regulating rate method, it is required for all thermal units in each province in the region to share peak regulating demands equally in accordance. In comparison, the model proposed in this paper focuses on the whole region, which can maximize the mutual aids of peak regulating resources in the whole regional power grid.

The calculation results of the two models are shown in Table 5. As can be seen from the comparison results, the total costs by using the equal peak regulating rate optimization model (model 2) are higher than the proposed model (model 1) in this paper. The new energy utilization rate is lower, and the solution time of the two models is relatively similar. As can be seen from the above comparison results, the proposed optimization model and method in this paper have higher calculation accuracy and solution efficiency. It can fully consider the mutual aid of peak regulating resources and can achieve the highest new

energy power utilization rate. What is more, it is more suitable for solving the actual optimization dispatching problems of regional power grids in China.

**Table 5.** Comparison of calculation results between two optimization models.

| Model | Optimization Results | |
| --- | --- | --- |
| | Total Costs/USD 100 Million | New Energy Utilization Rate% |
| 1 | 1.953 | 90.9 |
| 2 | 2.116 | 87.1 |

## 5. Conclusions

Based on deeply studying the peak regulation characteristics of different power supplies, an original two-layer optimal dispatching model is proposed in this paper. The upper layer is mainly the unit startup coordination and optimization scheduling model of each province in the region. The lower layer is mainly the optimal scheduling model of transmission curves in the region. By solving the two-layer optimization model, the lowest operation costs and the highest new energy utilization rate of the regional power grid can be obtained, which can realize full aid and sharing of peak regulating resources in the whole region. Finally, based on the simulation and analysis of the IEEE RTS-96 power system and an actual regional power grid, the feasibility and effectiveness of the proposed model and optimization method are verified.

**Author Contributions:** T.Y. established the two-layer optimization dispatching model of the regional power grids and established the simulation model for the verification. S.L. established the two-layer optimization dispatching model of the regional power grids. M.Z. corrected the manuscript and put forward many suggestions for improvement. Y.L. put forward some suggestions when building the model and analyzed the simulation results. W.F. checked and corrected the mathematical model of the paper. J.L. corrected the manuscript and put forward many suggestions for improvement. All authors have read and agreed to the published version of the manuscript.

**Funding:** The work was funded by the science and technology project guided by northeast branch of state grid corporation of China. Project Number: (52992623000G, SGDB0000DJJS2310060).

**Data Availability Statement:** Data is contained within the article.

**Conflicts of Interest:** Authors Tianmeng Yang, Yanchun Li, Wei Feng and Jicheng Liu were employed by the company Northeast Branch of State Grid Corporation of China; Meng Zhang was employed by the company China Energy Engineering Group Liaoning Electric Power Design Institute Co., Ltd. The remaining authors declare that the research was conducted in the absence of any commercial or financial relationships that could be construed as a potential conflict of interest.

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
