# Peer review of "Research on a Two-Layer Optimal Dispatching Method Considering the Mutual Aid of Peak Regulating Resources among Regional Power Grids"

_energies, doi:10.3390/en17030667_

Round 1
Reviewer 1 Report
Comments and Suggestions for Authors
The article presents an optimization proposed considering the mutual aids of peak regulation resources in order to determine the optimal units startup mode and power output in a China grid, thus obtaining the power supply output arrangements and the technical and economic indicators.
The paper needs improve its organization: english grammar review, spaces between paraghaps and section, units of the graphs, figures resolutions, variables descriptions (include a nomenclature list) and improve the conclusion.
The power system used to realize the study should be detailed, as well how it modeled and in which environment simulation were made.
The new contribution should be clarified in the introduction section, showed the bottlenecks of the literature the motivated this approach.
Comments on the Quality of English LanguageThe reviewer suggests to be attached a proof of english language review carried out by a specialized translation service.
Author Response
The authors are grateful to the reviewer’s time and efforts, which will help to improve the technical quality of this manuscript.
Comment 1:
The power system used to realize the study should be detailed, as well how it modeled and in which environment simulation were made.
Author response:
Dear editor, according to your opinion, the brief introductions of the power system and main solving tools of the case have been added in the article and is marked red.
The first power system in the case is an example system which is modified by IEEE RTS-96 example system, and all parameters of the system are based on the original system except load and wind power output curves.
The second power system in the case is modified by an actual regional power system in China. All the parameters of units and load curves are all based on the actual system. The typical load and new energy power output curves are clustered from the historical curves of a year.
The main simulation platform is based on Matlab and the optimization model is solved by CPLEX tools.
Author actions:
Modifications are marked in red in the section 3 of the revised paper.
Comment 2:
The new contribution should be clarified in the introduction section, showed the bottlenecks of the literature the motivated this approach.
The reviewer suggests to be attached a proof of English language review carried out by a specialized translation service.
Author response:
Dear editor, the main contributions of our work is to solve the unequal distribution problem between peak regulation requirements and resources in each provincial power grid. And to solve this problem, we put forward a two-layer optimization model which can maximize the distribution of peak regulation resources. Since there are few researches to study the optimization of peak regulation resources, all these contributions in the article are original. And all these contributions and originality are added in the abstract section of the article and have been marked red.
Author actions:
Modifications are marked in red in the abstract section of the revised paper.
Comment 3:
Comments on the Quality of English Language:
The new contribution should be clarified in the introduction section, showed the bottlenecks of the literature the motivated this approach.
Author response:
Dear editor, moderate editing of English language is done in this paper. And those complicated sentences have been optimized and become better understanding.
Author actions:
The whole paper has been revised in expression.

Reviewer 2 Report
Comments and Suggestions for Authors
This paper proposes a two-layer 16optimization dispatching model considering the mutual aids of peak regulation resources between each province. The reviewer puts forward the following comments to the authors:
1. Clearly highlight the technical contributions and originality of your work.
2. Have you considered the uncertainty of renewable energy? How does it impact the results of proposed model?
3. Is the proposed model scalable?
4. Please elaborate on the type of proposed optimization model, i.e. linear, non-linear?
Comments on the Quality of English LanguageModerate editing of English language required to enable better understanding. Some sentences are too long and complicated, such as "In [16-17], based on the Manson-Coffin formula, the loss costs model of thermal power units under variable load is established, and on the basis of full acceptance of wind power, the economic dispatching model of power system based on hierarchical deep peak regulation is established with the goal of minimizing the total power generation costs of generating units."
Author Response
The authors are grateful to the reviewer’s time and efforts, which will help to improve the technical quality of this manuscript.
Comment 1:
Clearly highlight the technical contributions and originality of your work.
Author response:
Dear editor, the main contribution of our work is to solve the unequal distribution problem between peak regulation requirements and resources in each provincial power grid. And to solve this problem, we put forward a two-layer optimization model which can optimize the distribution of peak regulation resources. Since there are few researches to study the optimization of peak regulation resources, all these contributions in the article are original. And all these contributions and originality are added in the abstract section of the article and have been marked red.
Author actions:
Modifications are marked in red in the abstract section of the revised paper.
Comment 2:
Have you considered the uncertainty of renewable energy? How does it impact the results of proposed model?
Author response:
Dear editor, the uncertainty of renewable energy are all considered in the model. We selected typical daily wind and PV power output scenarios by using clustering algorithm. Such as in the case study of the article, we selected four typical wind scenarios to represent all wind power output curves in a year. And the selected results of the typical renewable power output scenarios and typical scenario quantities may have different inflections to the optimization results. The more the typical wind scenarios’ quantity is, the more accurate the simulation results will be. Also ,the simulation speed will descend.
Author actions:
As explained above, the influence of uncertainty of renewable energy is added into the revised paper and are marked in red in Section 3.
Comment 3:
Is the proposed model scalable?
Author response:
Dear editor, the proposed model in the article is scalable. This optimization dispatching model can be applied into any regional power grid, which have AC power transmission exchanges between each power grid.
Author actions:
As explained above, the explanation of the model is scalabe is added into the revised paper and are marked in red in Section 2.1.
Comment 4:
Please elaborate on the type of proposed optimization model, i.e. linear, non-linear?
Author response:
Dear editor, the proposed model is non-linear. The variables should be optimized contain the startup modes and power output of each unit. The objective function of the optimization model is the total operation costs, the constraints of the model contain system power balance constraints, thermal power unit climbing constraints, pumped storage operation constraints and upper and lower limits of thermal power units. The objective function and constraints are all nonlinear functions. Therefore, the optimization model is nonlinear model.
Author actions:
As explained above, the introduction and solve tools of the optimization model are marked in red in Section 3.
Comment 5:
Comments on the Quality of English Language:
Moderate editing of English language required to enable better understanding. Some sentences are too long and complicated, such as "In [16-17], based on the Manson-Coffin formula, the loss costs model of thermal power units under
variable load is established, and on the basis of full acceptance of wind power, the economic dispatching model of power system based on hierarchical deep peak regulation is established with the goal of minimizing the total power generation costs of generating units."
Author response:
Dear editor, moderate editing of English language is done in this paper. And those complicated sentences have been optimized and become better understanding.
Author actions:
The whole paper has been revised in expression.

Round 2
Reviewer 1 Report
Comments and Suggestions for Authors
Thank you for the manuscript improvements. The paper contribution is well represented in the revised version.
Comments on the Quality of English Language
It was improved.
Author Response
The authors are grateful for the reviewer’s time and efforts, which will help to improve the technical quality of this manuscript.
Comment 1:
Minor editing of English language is required.
Author response:
Dear editor, moderate editing of English language has done carefully in this paper. And my manuscript has been checked by a colleague fluent in English writing.
Author actions:
The whole paper has been revised carefully in expression.

Reviewer 2 Report
Comments and Suggestions for Authors
The reviewer is satisfied with the revised draft and recommends its acceptance.
Author Response
The authors are grateful for the reviewer’s time and efforts, which will help to improve the technical quality of this manuscript.
Comment 1:
The reviewer is satisfied with the revised draft and recommends its acceptance.
Author response:
Dear editor, thanks a lot for your opinions of my article. We are so grateful for your time and efforts, which help to improve the technical quality of the manuscript.
Author actions:
Thanks a lot to your time and efforts for my article.
